# Spin-Induced Switching of Electronic State Populations in Transition Metal Polyphthalocyanines

**DOI:** 10.3390/ma15228098

**Published:** 2022-11-16

**Authors:** Deepali Jagga, Vitaly I. Korepanov, Daria M. Sedlovets, Artur Useinov

**Affiliations:** 1International College of Semiconductor Technology, National Yang Ming Chiao Tung University, Hsinchu 30010, Taiwan; 2Institute of Microelectronics Technology and High-Purity Materials, Russian Academy of Science, 142432 Chernogolovka, Russia

**Keywords:** polyphthalocyanines, magnetic configuration switching, resistance switching, spin-polarized states, metal-insulator transition

## Abstract

Polyphthalocyanines (PPCs) are a new and promising class of two dimensional materials offering versatile avenues for next generation electronic devices. For organic spintronic devices, PPCs can be engineered to tailor the electric and magnetic properties. In this work, we investigate PPC’s monolayers with embedded transition metal atoms (TM = Fe, Ni, Cu), utilizing first principle calculations based on spin-polarized generalized gradient approximation (SGGA). PPC sheets with central TM atoms are simulated for the dispersion curves, electronic density of states (DOS), and projected density of states (PDOS) using quantum atomistic toolkit (Quantum ATK) software. According to simulations, the FePPC supercell with four magnetic moments of Fe, aligned in a parallel ferromagnetic (FM) configuration, show the conductive FM state, while in the case of the anti-parallel antiferromagnetic (AFM) order of the magnetic moments, the material exhibits semiconducting non-magnetic behavior. FM-ordered NiPPC displays a metallic state, which is partly suppressed for AFM-ordered NiPPC. In contrast, non-magnetic CuPPC is found to be the best conductor due to its larger PDOS at the Fermi level among all considered systems.

## 1. Introduction

Two-dimensional materials have received much attention in the current technological era as the requisites for the ever-increasing miniaturization of integrated circuits. Growing industrial needs for devices with extended functionalities make it essential to look for novel materials with unique characteristics. Graphene, BN sheets, and TM dichalcogenides are a few classes of 2D materials which exhibit interesting properties, such as low spin-orbital coupling, high carrier mobility channels, small hyperfine interaction, and the freedom to alter the metallic surface states of topological insulators, especially at room temperature [1,2,3,4,5]. Among 2D materials, PPCs are promising well-known candidates for electronic applications. Their nitrogen-active sites are capable of redox reactions [6] and the presence of carbon sites effectively enhances the electron conductivity [7]. Chemically controlled pyrolysis of PPC films can be used to obtain two-dimensional films, similar to graphene. However, C, H, and N atoms in the PPC matrix are not spin-polarized. Fortunately, the ability to choose the magnetic TM dopants potentially facilitates the modulation of the spin-related magnetic properties of the material. The unique characteristics of PPCs are caused by the organic matrix that can incorporate various magnetic TM atoms, forming an ordered structure with a period of around 1.1 nm [8,9]. Possible magnetic switching between FM and AFM ordering of the neighboring atoms can potentially lead to variations in the material properties.

For example, Zhou et al. [8] found that PPCs with embedded TM atoms may act as semi-metals with FM or AFM properties. In their spin-polarized density functional theory (DFT) studies, positive and negative exchange-correlation energy indicated the FM and AFM behavior of TM-PPCs, respectively [8]. According to Y.Pei et al. [10], it is imperative to modulate the transport properties of the material as an efficient thermoelectric. Furthermore, the *I*-*V* characteristics show that TM-PPCs are promising materials with excellent tunneling, spin injection, and spin filtering capabilities [9,11,12,13]. Huang et al. [13] found highly appreciable spin filter efficiencies and a negative value of differential resistance using the transmission spectrum in FePPC. In general, TM-PPCs are important for applications such as oscillators, electronic amplifiers, switching devices [14,15] and memristors [16]. Wang et al. [12] studied the magnetic properties of PPCs with 5d TMs and found that PPCs are versatile materials that can be used for the evolution of memory-storage devices. Furthermore, energy control between the highest occupied and lowest unoccupied molecular orbitals (HOMO and LUMO) of CuPPC is suitable for enhanced photovoltaic performance [5,8] and the related power consumption efficiency [17,18,19,20,21].

The aim of the present study is to provide electronic and magnetic properties simulations for PPC monolayers with embedded Fe, Ni, and Cu TM atoms, using Quantum ATK software [22]. Computations and related electronic structure analyses are based on the DFT [21] within the framework of SGGA. It can help to provide semi-local approximations of the electronic correlations [23,24,25,26,27], taking into account Heisenberg exchange interactions.

## 2. Methods

The supercell of the TM (Fe, Ni, Cu)PPC monolayer contains four primitive cells with central TM atoms. Atomic position optimization for TM-PPCs supercells is achieved with an atomic force tolerance of 0.01 eV/Å for all systems. The Fritz Haber Institute (FHI) pseudo potential is selected as a basis for electron correlations, having a density mesh cut-off of 120 Hartree. The double-zeta polarized basis is implemented for all atoms in TM-PPCs. The band structure, DOS, and PDOS are derived in the range of the tetrahedron spectrum method with the related converged wave functions. The Monkhorst–Pack grid is used in the energy range from −1.0 eV to +1.0 eV for spin up and spin down electrons, accounting for atomic orbital contributions. The band structures are simulated according to SGGA along Γ, X and V symmetry points of the Brillouin zone, visualizing spin up and spin down orbital contributions. Dipole moment correction and spin polarization are considered in all calculations. All PPC systems were passed through the geometry relaxation procedure.

## 3. Results and Discussion

In this work, we simulated PPCs with embedded TMs (Fe, Ni, Cu) which have the electronic configurations of [Ar] $3d^{n}$$4s^{m}$, where *n* = 6, 8, *m* = 2 for Fe, Ni and *n* = 10, *m* = 1 for Cu, respectively. According to the crystal field theory, 3d-orbitals are characterized by degenerate energy states through the inter-atomic interactions in the square planer crystal field [8]. In addition to energy splitting, exchange coupling and spin-orbital interactions play a crucial role in the magnetic and transport properties of these materials. Quantum ATK shows the large transformation of the electronic states near the Fermi level due to switching between FM and AFM magnetic configurations for FePPC (Figure 1 and Figure 2) and NiPPC (Figure 3 and Figure 4), respectively. TM-PPCs display the absence of the band gap due to metallic states, as shown in Figure 1b, Figure 3b and Figure 4b, or a small band gap, demonstrating the semiconducting state, Figure 2b. In addition, it is found that the dispersion bands are spin split for the FM case, while for AFM they are not. The sharpest peak of the PDOS for the AFM configuration near $\varepsilon_{F}$ is an essential premise to observe good thermoelectric properties, as discussed in Ref. [28]. In contrast to FM ordering, AFM-ordered FePPC shows an indirect band gap, Figure 2b. Furthermore, there are flat bands along the X-V direction. These flat bands give rise to a higher effective mass, making them an ideal candidate for thermoelectric applications. A similar finding was reported in Ref. [28]. In addition, the corresponding PDOS (Figure 2d) confirms the semiconductor behavior, with a HOMO-LUMO gap E${}_{g}$, ranging in our case from −0.04 eV to +0.02 eV, that is in good agreement with Ref. [29]. For the FM ordering, the highest peaks of spin up and spin down PDOS contributions are observed near $\varepsilon_{F}$, revealing the existence of metallic states in FePPC (Figure 1d) and NiPPC (Figure 3d), respectively. However, for AFM-ordered NiPPC, there is an equal contribution from spin up and spin down PDOS at −0.45 eV, and zero PDOS in the range from −0.3 eV to −0.1 eV that clearly dictates the weak metallic characteristics (Figure 4d). Thus, FePPC stands out among other TM-PPCs reported here, having variable conducting properties due to FM to AFM switching. Additionally, FePPC was found to possess a high magnetic anisotropy, as reported in Ref. [30].

Since spin properties in TM-PPCs originate from the 3d-orbitals of the TM, the magnetism reflects the interaction between the orbitals of the metal atoms in the neighboring cells. The AFM configuration of FePPC, results in a finite band gap, and thus, exhibits semiconducting characteristics. In the case of CuPPC, the band structure was found to be different. Spin up and spin down states had the same contributions, as shown in Figure 5a. There are non spin-split concentrated peaks near $\varepsilon_{F}$ for PDOS, as shown in Figure 5b, which confirm the highly conductive (non-magnetic) behavior of CuPPC.

## 4. Conclusions

In this study, we found that initial magnetic moment configurations of the neighboring Fe atoms in PPC monolayers strongly modify their band structure, DOS, and PDOS. In the case of AFM-ordered FePPC, the simulation shows that the Fermi level is located in a narrow band gap, and all bands are not degenerated by spin. In contrast, FM-ordered FePPC band structures show spin-resolved bands, crossing the Fermi level. This reveals promising magnetic and conducting properties. Rapid magnetic and resistive changes from FM to AFM states make FePPC promising for various applications, including the field of quantum computations. The notable contribution of spin up or spin down components to PDOS near the Fermi energy level confirms the promising thermoelectric application for FM-ordered Fe- and NiPPCs. In contrast, for AFM-ordered Fe- and NiPPCs, the conductive nature is suppressed to a certain extent due to the absence of related PDOS peaks. Finally, the expected highest PDOS value at Fermi level is found for CuPPC. Therefore, these results clearly predict that CuPPC may have enhanced charge transport properties.

## Figures and Tables

**Figure 1 materials-15-08098-f001:**
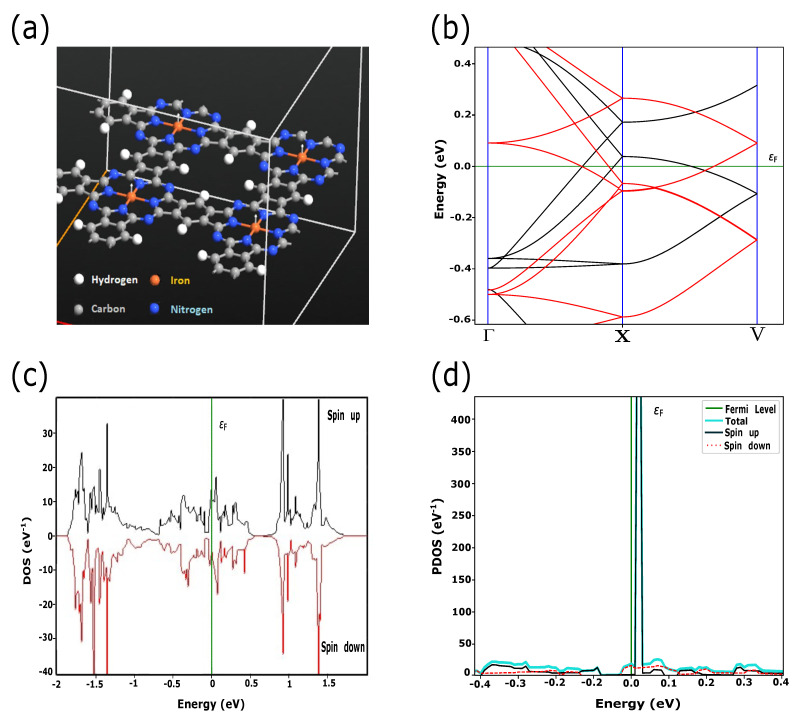
(**a**) FePPC supercell with parallel magnetic moments in Fe (white arrows), defined as FM ordered; (**b**) Spin up (black) and spin down (red) dispersion bands. (**c**) Spin-resolved DOS for FePPC; (**d**) PDOS, where the highest peak is formed due to spin up band contribution.

**Figure 2 materials-15-08098-f002:**
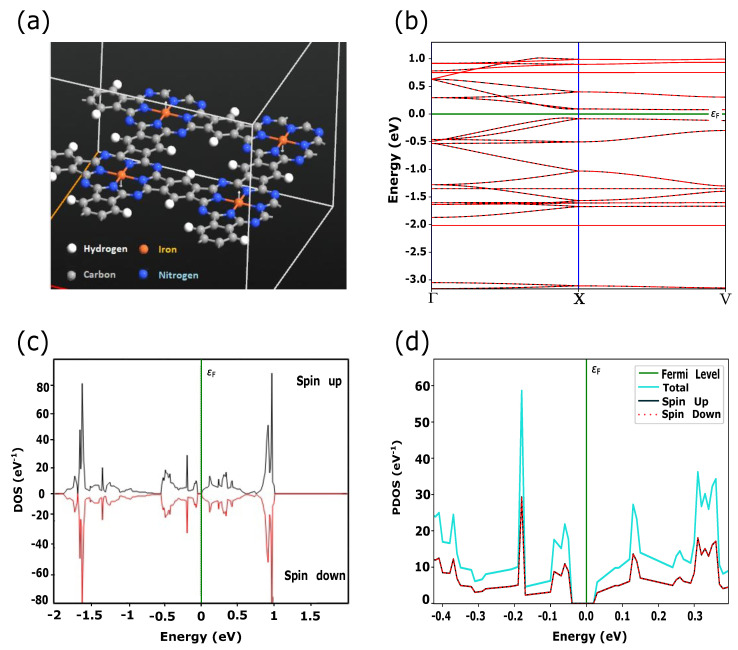
(**a**) FePPC with magnetic moments (white arrows) aligned in an anti-parallel configuration, defined as AFM ordered; (**b**) Dispersion bands, showing an indirect band gap along Γ-X-V symmetry points in *k*-space (in-plane PPC direction in real space); (**c**,**d**) DOS and PDOS, depicting the related band gap. Spin up and spin down components of electronic states are symmetrically equal.

**Figure 3 materials-15-08098-f003:**
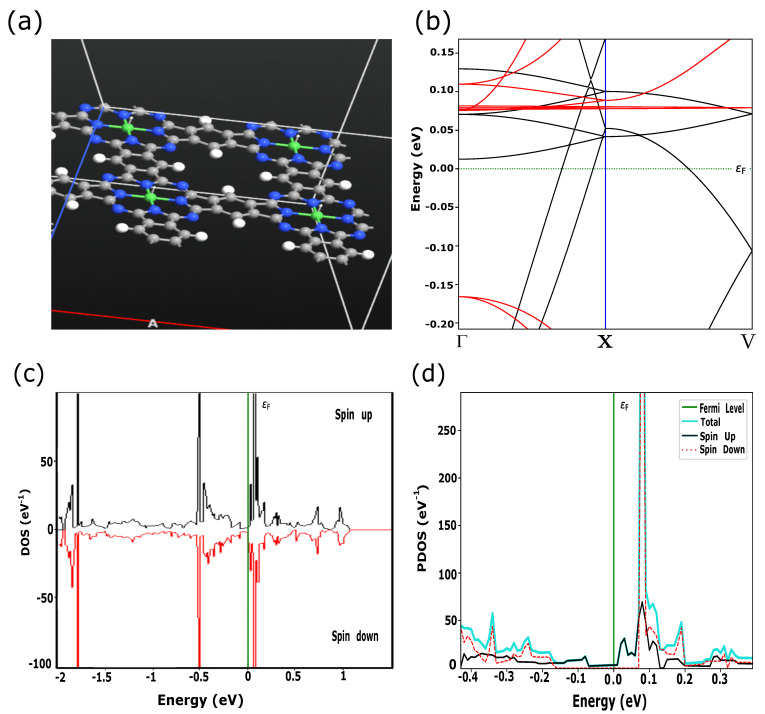
(**a**) NiPPC supercell with parallel magnetic moments in Ni (white arrows); (**b**) spin up (black) and spin down (red) dispersion bands. (**c**) Spin-resolved DOS for NiPPC; (**d**) PDOS, where the highest peak near $\varepsilon_{F}$ is related to spin down band contribution.

**Figure 4 materials-15-08098-f004:**
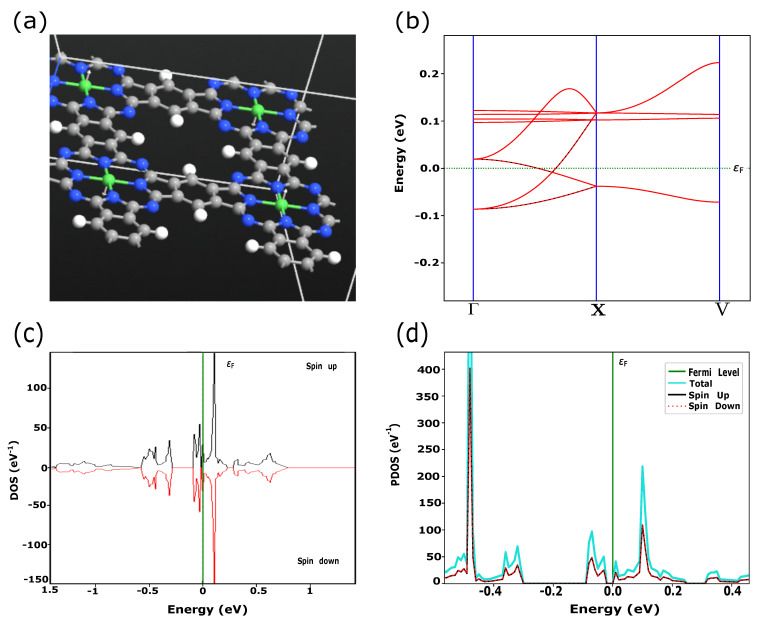
(**a**) NiPPC with magnetic moments (white arrows) aligned in anti-parallel configuration, determined as AFM in the text; (**b**) Dispersion bands; (**c**,**d**) DOS and PDOS depicting the weak metallic behavior.

**Figure 5 materials-15-08098-f005:**
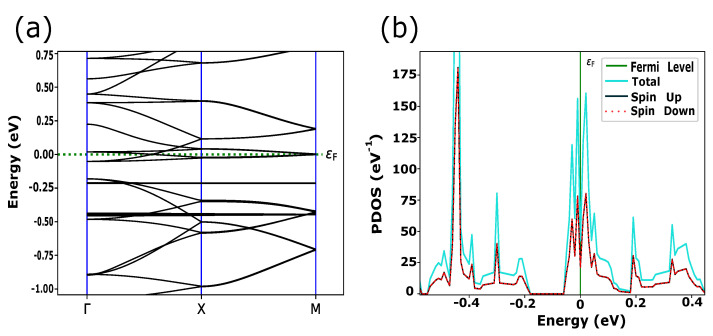
(**a**) Band structure in CuPPC (**b**) PDOS for CuPPC.

## Data Availability

Not applicable.

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
