# Peer review of "Spin-Induced Switching of Electronic State Populations in Transition Metal Polyphthalocyanines"

_materials, 2022, doi:10.3390/ma15228098_

Round 1
Reviewer 1 Report
The authors describe the results of their band structure calculations on polyphthalocyanines including the metal free version (described as ‘pristine’ by the authors) along with Fe and Cu substituted analogs. Since Fe phthalocyanine is already known to have an unusual electronic structure (an intermediate spin S=1 state) the results of thece calculations could be quite interesting and may provide guidance for the synthesis of novel materials. However some refinement of this paper is needed before publication.
1. The authors should clarify what is meant by ‘pristine’. Phthalocyanine itself includes two hydrogen atoms coordinated in the central ring. Did the authors computational model include the equivalent hydrogens in the polymeric model?
2. In the second paragraph of the results and discussion the authors refer to ‘octahedrally coordinated sites’ but the coordination sites in pthalocyanines and related molecules are quite clearly square planar. This could be a simple typo, but the authors then continue to discuss d orbital splitting in an octahedral field which is confusing and possibly not even relevant. In fact the uniquely interesting electronic structure of iron phthalocyanine results from it *not* being octahedral.
The language in this paragraph is, in general rather confusing. What is meant by ‘degeneration of spin polarized states due to possible influence of atomic moments or condition of the d orbital overlapping’ - by atomic moments are they referring to the magnetic moments from the Fe d electrons? Considering that the half filled orbitals in FePc (that give rise to the magnetic moment) have significant pi character, how localized are they on the metal atom? What is meant by ‘d-orbital overlapping’ - overlapping with which other orbitals?
3. What do the authors mean by a ‘valuable’ band gap?
4. In general, the English in the paper could use significant ‘polish’
Reviewer 2 Report
The authors model atomic configurations and electronic structure of two-dimensional materials polyphthalocyanines ( PPCs), with and without transition metal atoms using a quantum atomistic toolkit ( ATK) software. It is a tool that is now widely used to design novel materials with best desired properties. ATK incorporates first principles band structure calculations based on spin-polarized generalized gradient approximation ( SGGA). According to simulations Fe-PPC monolayer with ferromagnetically ordered Fe moments is metallic, whereas for anti-parallel alignment the material exhibits insulating non-magnetic behavior. It is expected that Cu-PCC to be a well conductive material.
Due to the use of such an excellent tool as ATK, I do not know how creative the authors' achievements are. Certainly PPC is a technologically important material, especially for spintronics and in this respect it is important to publish the present results. Interesting is that possible magnetic switching between differently magnetically ordered states can lead to radical change of transport properties.
In my opinion the study adds new interesting knowledge to the field and can serve as a hint in searching new potential application of phthalocyanines. The paper deserves for publication, however, it would be necessary to relate the calculated values, e.g. the width of the gap to the experimental studies of the analyzed material or similar systems. Although embedding PPC with iron proves to be interesting, a comment on material engineering with other transition atoms would be useful.
The paper is well written and the obtained results are important to design novel materials for spintronics and therefore, after applying the above note, I recommend publication of this manuscript without modifications.
Author Response
Many thanks to the Reviewer for the positive feedback.
Round 2
Reviewer 1 Report
The authors have adequately addressed my previous comments; the paper is now suitable for publication